# Hydroxychloroquine Mitigates Dilated Cardiomyopathy Phenotype in Transgenic D94A Mice

**DOI:** 10.3390/ijms232415589

**Published:** 2022-12-09

**Authors:** Rosemeire M. Kanashiro-Takeuchi, Katarzyna Kazmierczak, Jingsheng Liang, Lauro M. Takeuchi, Yoel H. Sitbon, Danuta Szczesna-Cordary

**Affiliations:** 1Department of Molecular and Cellular Pharmacology, University of Miami Miller School of Medicine, Miami, FL 33136, USA; 2Interdisciplinary Stem Cell Institute, University of Miami Miller School of Medicine, Miami, FL 33136, USA

**Keywords:** dilated cardiomyopathy (DCM), echocardiography, hydroxychloroquine (HCQ), MYL2 gene, regulatory light chain (RLC), super-relaxed state, transgenic mice

## Abstract

In this study, we aimed to investigate whether short-term and low-dose treatment with hydroxychloroquine (HCQ), an antimalarial drug, can modulate heart function in a preclinical model of dilated cardiomyopathy (DCM) expressing the D94A mutation in cardiac myosin regulatory light chain (RLC) compared with healthy non-transgenic (NTg) littermates. Increased interest in HCQ came with the COVID-19 pandemic, but the risk of cardiotoxic side effects of HCQ raised concerns, especially in patients with an underlying heart condition, e.g., cardiomyopathy. Effects of HCQ treatment vs. placebo (H_2_O), administered in Tg-D94A vs. NTg mice over one month, were studied by echocardiography and muscle contractile mechanics. Global longitudinal strain analysis showed the HCQ-mediated improvement in heart performance in DCM mice. At the molecular level, HCQ promoted the switch from myosin’s super-relaxed (SRX) to disordered relaxed (DRX) state in DCM-D94A hearts. This result indicated more myosin cross-bridges exiting a hypocontractile SRX-OFF state and assuming the DRX-ON state, thus potentially enhancing myosin motor function in DCM mice. This bottom-up investigation of the pharmacological use of HCQ at the level of myosin molecules, muscle fibers, and whole hearts provides novel insights into mechanisms by which HCQ therapy mitigates some abnormal phenotypes in DCM-D94A mice and causes no harm in healthy NTg hearts.

## 1. Introduction

Heart failure (HF) has emerged as a critical global health concern and searching for efficient prevention and therapeutics is one of the main goals of current worldwide research. Hydroxychloroquine (HCQ), a 4-aminoquinoline agent used for more than 50 years to prevent/treat malarial infections, resurfaced recently as a potential first-line pharmacotherapy in the face of ongoing COVID-19 epidemics [1]. Yet, the use of HCQ was associated with adverse cardiac effects, e.g., long QT syndrome [2], and prolonged exposure to HCQ was also related to the development of restrictive or dilated cardiomyopathy (DCM) [3]. Development of cardiomyopathy, however, was observed after decades of HCQ treatment of patients [4,5]. The acute effect of HCQ on the progression of cardiovascular phenotype, especially in individuals suffering from inherited cardiac diseases, e.g., genetic DCM, has never been fully characterized, posing the question of the usefulness of HCQ pharmacotherapy.

This study aimed to test whether short-term and low-dose treatment with HCQ can modulate heart performance in a preclinical model of DCM compared to healthy hearts of non-transgenic (NTg) littermates. The study on NTg animals was to test whether HCQ treatment can trigger the development of cardiomyopathy and on transgenic (Tg) DCM mice to determine if the phenotype would worsen on the administration of HCQ. We employed a previously created mouse model of DCM associated with the Aspartate94-to-Alanine94 mutation in the regulatory light chain (RLC) of myosin encoded by the MYL2 gene [6,7]. Hearts of the model express the D94A-RLC mutation that was identified in the DCM cohort, adding, for the first time, the MYL2 gene to the list of DCM target genes [8].

DCM can be characterized by progressive left ventricular (LV) dilation, low ejection fraction (EF), and impaired LV contraction leading to systolic dysfunction and HF [9]. It is estimated that 25–48% of all DCM cases originate from gene defects [10], many of which are mutations in sarcomeric proteins [11]. Our previous study on Tg-D94A mice convincingly showed a close resemblance of the phenotype observed in mice to human DCM [6]. Consistent with the DCM phenotype, Tg-D94A mice showed significantly reduced EF and fractional shortening (FS), decreased cardiac output (CO) and increased LV cavity dimensions in systole and diastole, as well as increased end-diastolic/systolic volumes [6]. Histopathology evaluation of LV tissue from ~5- and ~12-mo-old D94A animals showed no apparent myofilament disarray or fibrosis; still, the rare fibrotic depositions occurred in the hearts of older male but not female D94A mice [6].

In the current investigation, we have assessed the effect of a short-term HCQ treatment on myosin motor function and heart performance using male and female Tg-D94A mice compared with sex-matched NTg littermates. Both groups of mice received a low-dose HCQ administered in drinking water for one month (Figure 1), followed by in vivo and in vitro evaluations of heart function at the level of myosin molecules, skinned muscle fibers, and whole hearts. The main goal was to determine the physiological crosstalk between HCQ treatment and cardiovascular responses in healthy NTg mice and the Tg-D94A model of DCM. Our data suggest that short-term treatment with HCQ produces no harmful effects on heart function in either NTg or Tg-D94A mice. On the contrary, short-term and low-dose HCQ application mitigated some aspects of cardiac dysfunction in the DCM myocardium, suggesting a need for additional therapeutic investigations in the future.

## 2. Results

In this study, we investigated whether administering a low dose of HCQ into genetically altered mice for one month would be beneficial or detrimental to the overall health and cardiac function of Tg-D94A, a mouse model of DCM [6]. Because of research reports implicating the involvement of HCQ treatment in the development of cardiomyopathy in patients [4,5,12], we assessed the heart function in healthy NTg littermates treated with HCQ vs. placebo (H_2_O). Experiments included measurements of heart function at all levels of system organization, from myosin molecules, myofilaments, and skinned muscle fibers to the whole heart, using our established state-of-the-art analyses [6,7,13].

Figure 1 shows the experimental design and provides a list of assays performed to evaluate the effects of HCQ on myosin motor function and heart performance in male (M) and female (F) DCM mice compared with NTg littermates. Figure 2 depicts the daily consumption of water ± HCQ for all mice over one month. On average, males consumed statistically more liquid per day (~5 mL) compared with females (~4 mL) (Figure 2A), but no differences in liquid consumption were noted between HCQ vs. placebo (H_2_O) for both groups of mice (Figure 2B). Despite larger liquid consumption by male mice, we have not observed sex differences in the experiments listed in Figure 1.

### 2.1. Effect of HCQ Administration on the in vivo Heart Function

For simplicity, Table 1 and Appendix A report the data at baseline and 30 days of treatment. Echocardiography evaluation of cardiac morphology at baseline and then on days 10, 20, and 30 of treatment revealed no HCQ-induced changes in left ventricular inner diameter in systole (s) and diastole (d) (LVIDs,d) and anterior or posterior wall thickness (LVAWs,d, LVPWs,d) in any of the groups (Table 1). Similarly, no significant changes were observed in chamber volumes, i.e., end-diastolic or systolic volumes (EDV, ESV), with the heart rate (HR) of mice maintained at ~540 bpm (Table 1).

Global longitudinal strain (GLS) analysis, conducted as described in Dulce et al. [14], showed markedly reduced GLS (absolute values) in Tg-D94A mice in comparison to NTg animals at baseline and 30 days after treatment with HCQ (Figure 3A). Interestingly, Tg-D94A mice showed a significant improvement in GLS after 30-day treatment with HCQ, and the difference between not treated vs. HCQ- treated Tg-D94A mice was statistically significant (Figure 3B, *p* < 0.01). NTg hearts were not affected by HCQ treatment, and GLS values were similar between all NTg groups (Table 1). Non-invasive Doppler measurements also showed changes in the tricuspid annular plane systolic excursion (TAPSE), determined to evaluate the systolic function of the right ventricle (RV). At baseline, a significant decrease in TAPSE was observed in Tg-D94A mice vs. NTg mice, suggesting the possibility of RV systolic dysfunction in the mutant (Table 1). Notably, HCQ was able to restore TAPSE values in Tg-D94A mice after 30 days of HCQ treatment (Table 1).

Invasive hemodynamics evaluation did not reveal any major changes in cardiac function in HCQ-treated Tg-D94A or NTg mice (Appendix A). Small but significant differences were observed in the ratio between arterial elastance (Ea) and slope of end-systolic pressure-volume relationship (slope of ESPVR or Ees) (Ea/Ees) between placebo-treated Tg-D94A vs. NTg mice (Appendix A). Ees represents the cardiac contractility, and Ea denotes arterial load that conveys all the extracardiac forces opposing ventricular ejection [15]. Augmented Ea/Ees in Tg-D94A mice returned to basal levels following 30 days of HCQ treatment (Appendix A).

### 2.2. Effect of HCQ Treatment on Gross Morphology, Histopathology, and Ultrastructure

The gross morphology of 6-mo-old female Tg-D94A and NTg hearts reveals a visibly larger size of the mutant heart than NTg, arguably reflecting the dilated phenotype in Tg-D94A mice (Figure 4A). However, morphometric data showed no difference in heart weight/tibia length, the weight of atria, or water content in the lungs between Tg-D94A and NTg mice, indicating no differences between the HCQ- vs. placebo treatment (Figure 4B). Histopathology and transmission electron microscopy (TEM) evaluations of LV heart samples from HCQ treated 6-mo-old females showed no signs of myofilament disarray or fibrosis (Figure 5A); still some disrupted sarcomere structures and abnormal vacuolar formations were observed in Tg-D94A myocardium compared with NTg mice (Figure 5B). Quantification of fibrosis was performed by hydroxyproline (HOP) assay (Figure 5C), as described previously [16]. No differences between the groups or treatments were noted (Figure 5C).

Due to reports of HCQ-induced mitochondrial fragmentation [17,18], TEM was used to evaluate LV samples from mice (at 1000×, 3000×, and 5000× magnification) for potential mitochondrial abnormalities. The measurement of the average area of intermyofibrillar mitochondria (IFM) at 3000× for Tg-D94A vs. NTg ±HCQ showed no differences in IFM between the groups (Figure 5D), suggesting that short-term and low-dose HCQ treatment of mice does not result in mitochondrial damage.

### 2.3. Gene Expression Profiles in HCQ- vs. Placebo-Treated Tg-D94A and NTg Hearts

Expression of Angiotensin-converting enzyme 2 (ACE2), Atrial Natriuretic Factor (ANF), Collagen type VIII, alpha I (Col8a1), and B-type Natriuretic Peptide (BNP) was assessed in the hearts of HCQ- vs. placebo-treated Tg-D94A and NTg mice with Glyceraldehyde-3-phosphate dehydrogenase (GAPDH) used as a housekeeping gene (Figure 6). The data are presented as relative expression normalized to a placebo-treated male NTg mouse (expression = 1). The results for ACE2 transcript, which is a target receptor for SARS-CoV-2, revealed approximately ~1.5-fold lower expression of ACE2 in the hearts of HCQ-treated compared with placebo-treated NTg mice (Figure 6, ^&^
*p* < 0.05). Since the HCQ resurfaced during the COVID-19 pandemic as a first-line pharmacotherapy, we measured the expression of ACE2, shown to function as a host cellular entry receptor for coronavirus that directly binds the viral spike protein [19]. ACE2 has also been found to be crucial for maintaining normal cardiovascular functions and is highly expressed in human heart failure [20]. Therefore, its downregulation upon HCQ treatment (Figure 6) can be considered beneficial and suggests that HCQ may protect normal NTg hearts against virus entry into the cells and against cardiac injury.

ANF and BNP peptides are closely related to LV function and are sensitive markers of the severity of heart injury [21,22] and sensitive markers of pathological DCM phenotype [23]. BNP and its N-terminal (NT)-prohormone (NT-proBNP) are released in response to changes in pressure inside the heart, and their levels were seen to increase in patients with heart failure [24,25]. A high plasma concentration of NT-proBNP was observed to be indicative of an increased risk of cardiovascular mortality in elderly people [24]. Interestingly, treatment with HCQ led to a lower value of BNP expression in Tg-D94A vs. HCQ-treated NTg hearts (Figure 6). Therefore, HCQ therapy can potentially benefit Tg-D94A hearts that show lower BNP levels compared with HCQ-treated NTg animals (Figure 6).

On the other hand, no changes in the expression of ANF and Collagen VIII between HCQ- and placebo-treated groups were noted (Figure 6). ANF is upregulated in both ischemic and idiopathic cardiomyopathies [23], while collagen VIII may have a potential role in DCM development, especially the progression of cardiac remodeling and fibrosis [26]. Unchanged expression of both ANF and ColVIII suggests that no pathological remodeling occurred in the myocardium of Tg-D94A or NTg mice due to HCQ treatment. This result is in accord with the lack of fibrosis determined in all examined mouse groups by HOP assay (Figure 5C).

### 2.4. Contractile Function in Skinned Papillary Muscle Strips from HCQ vs. Placebo-Treated Tg-D94A and NTg Mice

Although the collective safety profiles of HCQ are relatively favorable, the chronic use of HCQ and its effect on the cardiovascular health of patients, especially those with an underlying heart condition, e.g., genetic cardiomyopathy, is not known [27]. To gain insight into the physiological effects of HCQ on the contractile properties of cardiac muscle and the ability of myosin motors to interact with actin and produce force, skinned LVPM fibers were tested for isometric force development over a full range of calcium concentrations, from relaxing pCa 8 to full activation at pCa 4, and sarcomere length (SL) ~2.1 µm [7]. The data showed no HCQ-mediated changes in contractile function in Tg-D94A and NTg mice (Figure 7A,B). The values of maximal force per cross-section of muscle strip, pCa_50_, depicting Ca^2+^ sensitivity of force, and Hill coefficient (n_H_) were not significantly affected in response to HCQ treatment for DCM and NTg mice (Table 2).

### 2.5. The Effect of HCQ on the Super-Relaxed State of Myosin in HCQ vs. Placebo-Treated Tg-D94A and NTg Mice

The heart’s contraction is a tightly regulated process that depends on the Ca^2+^ and ATP-dependent interaction between the actin-containing thin filaments and the myosin-containing thick filaments. During cardiac muscle contraction/relaxation, myosin can be characterized by three states: an active/force-producing state, a disordered relaxed state (DRX), and a super-relaxed state (SRX) [28]. In DRX, myosin cross-bridges protrude into the interfilament space but are restricted from binding to actin and producing force. In SRX, they display an ordered head arrangement along the thick filament axis and a highly inhibited ATP turnover rate [29]. The presence of the folded state of myosin, where the heads interact with each other and with the part of myosin heavy chain (MHC) and form so-called interacting head motif (IHM) structures, is hypothesized to be the origin of the SRX [29].

Here, we assessed the effect of HCQ on the SRX state of myosin and SRX↔DRX equilibrium in the hearts of mice using a single nucleotide exchange method in skinned LVPM fibers from Tg-D94A and NTg littermates (Figure 8). Fluorescence decay curves vs. time were collected on the exchange of fluorescent mant-ATP for non-labeled ATP (Figure 8A). The data were fitted to a two-state exponential equation to derive the amplitudes of the fast (P1) and slow (P2) phases of fluorescence decay and their respective T1 and T2 lifetimes (in seconds) [30,31]. To estimate the number of myosin heads that directly occupy the DRX vs. SRX states (Figure 8B), the rapid phase of the fluorescence decay (P1) was corrected for the fast release of nonspecifically bound mant-ATP, as described in Hooijman et al. [28]. The percent of myosin heads in the DRX vs. SRX states was then calculated using a correction factor of 0.44 for nonspecific mant-ATP binding, and the number of heads in the SRX state was estimated as P2/(1-0.44) [7]. Our results indicate that in the DCM Tg-D94A mouse model, the HCQ promotes the switch from the energy-conserving SRX state into the DRX state (Figure 8C). The SRX-to-DRX transition in the DCM Tg-D94A hearts is presumed to activate more cross-bridges to interact with actin and produce force (Figure 8C). This effect of HCQ is considered potentially beneficial for the Tg-D94A hearts, which show, consistent with DCM, hypocontractile phenotype [6]. It is anticipated that HCQ may improve the function of myosin motors and, thus, cardiac muscle contraction in DCM-D94A mice. No effect on SRX↔DRX equilibrium was observed in NTg mice (Figure 8, Table 3).

## 3. Discussion

In this report, we aimed to evaluate the effects of short-term and low-dose HCQ treatment on heart function in a preclinical model of DCM associated with the MYL2 gene [6,7] and compare the results with those of HCQ-treated NTg littermates. Conventional echocardiography did not reveal HCQ-induced cardiac dysfunction or detrimental changes in cardiac morphology in Tg-D94A or NTg mice (Table 1). On the contrary, the HCQ treatment resulted in significant improvement of some electrophysiological parameters in the Tg-D94A model of DCM. Even though the model has not shown as strong of the phenotype as we published previously [6], the assessment of GLS by speckle-tracking echocardiography detected an impaired cardiac performance in Tg-D94A mice at baseline and on day 30 of HCQ treatment (Figure 3A, Table 1). Echocardiographic global longitudinal strain is widely recognized as a more effective technique than conventional ejection fraction in detecting subtle changes in LV function [32], and it was significantly altered in Tg-D94A vs. NTg mice at baseline, indicating poorer myofilament shortening in the DCM model. 30-day treatment with HCQ significantly improved GLS in Tg-D94A mice (Figure 3B, *p* < 0.01 for Tg-D94A not treated vs. HCQ-treated), while no response to HCQ treatment was observed in NTg mice (Figure 3B). Pressure-Volume loops examination showed significant differences in the ratio between arterial elastance and slope of ESPVR between placebo-treated Tg-D94A vs. NTg mice, and the augmented Ea/Ees ratio observed in Tg-D94A mice returned to basal levels following 30 days of HCQ treatment (Appendix A).

Non-invasive Doppler measurements indicated changes in the tricuspid annular plane systolic excursion with a significant decrease in TAPSE observed in Tg-D94A mice vs. NTg mice at baseline (Table 1). This result suggested the altered systolic function of the right ventricle in DCM mice, and the data are consistent with those obtained in a mouse model of hypoxia-induced pulmonary hypertension reported by Zhongkai Zhu et al. [33]. Interestingly, 30-day treatment with HCQ restored TAPSE values in Tg-D94A compared with NTg mice (Table 1). This is an important finding with clinical implication suggesting that HCQ could be used therapeutically to alleviate the results of hypoxia on the systolic function of the right ventricle in DCM Tg-D94A mice [33]. HCQ-related improvement in GLS in the DCM model also suggests that the drug may be used clinically to correct subtle defects in LV function in DCM subjects.

Several earlier investigations have demonstrated that pharmacological preconditioning with HCQ may induce a cardioprotective effect against heart injury. Studies in neonatal rat cardiomyocytes treated with 2000 ng/mL HCQ and exposed to ischemia/reperfusion injury showed protective effects of HCQ during the reperfusion stage [34]. The authors speculated that enhancement of phosphorylation of the pro-survival kinase ERK1/2 was underlying HCQ-induced cardiac protection [34]. Another study also suggested that HCQ may enhance endothelial ERK5 phosphorylation, thereby exerting vasoprotective properties [35]. HCQ has also been suggested to act as a bradycardia agent with the potential for treating ischemic heart disease and heart failure [36]. Interestingly, reports of HCQ rendering cardioprotective and not cardiotoxic effects were associated with a short-term drug delivery [4].

Evaluation of LV tissue of 6-mo-old female mice for morphology, histopathology, and TEM ultrastructure showed mild effects of HCQ treatment on myofilament ultrastructure in Tg-D94A mice (Figure 5B) and no occurrences of fibrosis (Figure 5C). Likewise, the contractile function of the hearts tested in LVPM fibers of Tg-D94A and NTg mice remained unchanged following 30 days of HCQ treatment (Figure 7 and Table 2).

At the molecular level, we tested whether the SRX state of myosin is affected by HCQ treatment in Tg-D94A vs. NTg mice compared with the placebo-treated animals. Under relaxation conditions, myosin heads exist in various structural and biochemical states, and each state is associated with different energy consumption rates [9]. Since the resting phase of the cardiac cycle is essential for normal heart function, the unbalanced SRX↔DRX equilibrium may be indicative of an energetically compromised sarcomere. Our results show that pharmacological treatment with HCQ promoted the switch from the energy-conserving SRX state into the DRX state in the DCM Tg-D94A mouse model but not in NTg treated fibers, where no effect on SRX↔DRX equilibrium was observed (Figure 8, Table 3). As we reported previously, the Tg-D94A model of DCM was associated with systolic dysfunction [6], which was confirmed by GLS measurements (Figure 3A). Therefore, the HCQ-triggered SRX-to-DRX transition in Tg-D94A mice may be seen as fine-tuning of cardiac sarcomeres to increase the proportion of myosin heads readily available to interact with actin and produce force, leading to improved muscle contraction in DCM Tg-D94A mice.

This bottom-up investigation, from myosin molecules and muscle fibers to intact hearts, provides insights into short-term and low-dose HCQ therapy in a mouse model of DCM vs. NTg littermates. Based on echocardiography, invasive hemodynamics, histopathological/TEM, and muscle contractile mechanics/energetics evaluations, we can conclude that HCQ exerts no harmful effects on the hearts of NTg or DCM mice. On the contrary, 30-day treatment with low-dose HCQ significantly improved GLS and sarcomeric shortening in DCM-D94A hearts. Our collective results at the level of myosin motors and whole hearts suggest a need for additional investigations of the potential therapeutic benefits of HCQ in DCM subjects in the future.

## 4. Materials and Methods

### 4.1. Transgenic Mice

The generation and characterization of Tg-D94A mice used in this study, including transgenic D94A protein expression profiles, have been described earlier [6]. Two lines of six- to nine-mo-old male and female Tg-D94A mice were used in HCQ experiments, line 1 (L1), expressing 53.4 ± 3.2%, and line 2 (L2), expressing 49.5 ± 2.8% of D94A mutant in mouse hearts and the results were compared to sex-matched non-transgenic (NTg) littermates [6]. The percent of D94A expression in Tg-D94A L1 and L2 mice was confirmed in the current study using Western blots labeled with RLC-specific polyclonal CT-1 antibodies produced in this lab [37]. Myofibrils isolated from the LV of Tg-D94A L1 and L2 and control NTg hearts were run on 2D SDS-PAGE. Transgenic human ventricular RLC-D94A was separated from endogenous mouse ventricular RLC by their different pI points (Appendix A). Experimental details and the percent of D94A protein expression in Tg-D94A L1 and L2 are presented in the Appendix A.

### 4.2. Experimental Protocol and Treatment with Hydroxychloroquine (HCQ)

The effect of HCQ vs. placebo (H_2_O) on cardiac function, morphology, and myosin molecular characteristics was tested in 5–8-mo-old Tg-D94A and NTg mice of both sexes. Both groups of mice received a low dose of approximately 10 mg/Kg of HCQ (Spectrum Laboratory Products Inc., Gardena, CA, USA) [38], administered daily in drinking water for one month (Figure 1 and Figure 2). Electrocardiogram and echocardiography measurements were performed at baseline and then on days 10, 20, and 30 of HCQ administration. At the end of the experiment, both groups of mice underwent the invasive Pressure-Volume Loop hemodynamic measurements. The mouse hearts were then harvested and evaluated for gross morphology, histopathology, and ultrastructure. For functional studies, we used chemically skinned left ventricular papillary muscle (LVPM) fibers from mice and measured the steady-state force development and the effect of HCQ on the myosin super-relaxed (SRX) state. The ultimate goal was to assess the physiological crosstalk between HCQ treatment and cardiovascular responses in healthy NTg mice and the Tg-D94A model of DCM.

### 4.3. In Vivo Assessment of Cardiac Function

#### 4.3.1. Echocardiography

In vivo cardiac morphology and function of HCQ vs. placebo (drinking H_2_O)-treated Tg-D94A and NTg mice were assessed with a Vevo 2100 (Visual Sonics, Toronto, ON, Canada) equipped with an MS400 transducer as described previously [6,39]. Heart images were recorded from mice under isoflurane (Vedco Inc., Saint Joseph, MO, USA) inhalation anesthesia (1–2%) with monitored heart rates (above 500 beats/min) and body temperature maintained at 37 °C. M-mode and B-mode images were evaluated using AutoLV analysis software (Vevo LAB 5.6.1, FUJIFILM, Visual Sonics, Toronto, ON, Canada). Determined parameters from the M-mode images included LV end-diastolic (d) and end-systolic (s) dimensions (LVIDd,s), posterior and anterior wall thickness (LVPWd,s & LVAWd,s), and fractional shortening (FS). The LVVd,s (end-diastolic/systolic volumes) and ejection fraction (EF) were calculated from B-mode long-axis parasternal views. Mitral pulse-wave and tissue Doppler acquisitions were used to assess cardiac conditions during diastole. Tricuspid annular plane systolic excursion (TAPSE) and pulmonary pulse-wave were measured to assess right ventricular (RV) function. Global longitudinal strain (GLS) analysis was conducted using a workstation dongle #50794, Vevostrain TM software version 1.4 (Visual Sonics, Toronto, ON, Canada), as we described in Dulce et al. [14]. Briefly, GLS was calculated using B-mode images that were acquired from the parasternal longitudinal axis view. Three consecutive cardiac cycles were selected for analysis by semi-automated tracing of the endocardial and epicardial borders.

#### 4.3.2. Invasive Hemodynamics

The hemodynamic assessments and pressure-volume loops analysis were performed to assess the load-independent measures of ventricular systolic and diastolic function, as described earlier [6,39]. Mice were anesthetized in a chamber saturated with isoflurane (3%) and transferred to a surgical bench where anesthesia was maintained with ~2% isoflurane via an endotracheal tube, and the body temperature was controlled at 37 °C. During the procedure, a 6% albumin solution (Grifols Biologicals, LLC., Los Angeles, CA, USA) was infused into the jugular vein at the rate of 5 µL/min. The micro-tip catheter transducer (SPR-839; Millar Instruments, Houston, TX, USA) was introduced into the left ventricle through the right carotid artery. The LV pressure-volume loops were recorded at steady-state and during inferior vena cava occlusion. The volume was calibrated by echocardiographic measurement of the end-diastolic volume and stroke volume (SV). Diastolic performance was assessed by the measurement of the peak rate of LV relaxation (−dP/dt_min_), end-diastolic P–V relationship, and the time constant of LV relaxation (tau in ms). Cardiac preload was indexed as the EDV and end-diastolic pressure (EDP). Cardiac afterload was evaluated as effective arterial elastance (Ea): ESP/SV. Myocardial contractility was indexed by the peak rate of rise in LV pressure (dP/d_tmax_) and the load-independent end-systolic elastance (Ees), which is the slope of the end-systolic pressure-volume relationship (ESPVR). All analyses were performed using LabChart Pro software version 8.1.5, ADInstruments, Colorado Springs, CO, USA).

### 4.4. Morphometric, Histological, and Ultrastructure Assessments

At the end of the study, mice were euthanized by inhalation of isoflurane followed by thoracotomy, and standard morphometric measures were obtained, including body, heart, ventricles, atria, and lung (wet to dry) weights and tibia length. Lung wet weight was first determined. The lung tissue was then dried at 37 °C to a constant weight, and then the dry weight of the lung tissue was determined. The relative water content of lung tissue was calculated using the following equation: Lung water content = (wet lung weight − dry lung weight)/wet lung weight × 100%.

#### 4.4.1. Histological Assessment

After euthanasia, the hearts of representative 6-mo-old HCQ or placebo (H_2_O)-treated female Tg-D94A and NTg mice were excised, rinsed with PBS, weighed, and immersed in 10% buffered formalin overnight. Hearts were sectioned longitudinally, and the posterior parts (dedicated for histology analysis) were paraffin embedded. Slides were stained with hematoxylin and eosin (H&E) and Masson′s trichrome (prepared by Histology Laboratory, University of Miami Miller School of Medicine). Gross morphology and the presence and degree of fibrosis were evaluated using a Dialux20 microscope (Ernst Leitz Ltd., Midland, ON, Canada), 40×/0.65 NA (numerical aperture) Leitz Wetzlar objective, and an AxioCam HRc (Carl Zeiss MicroImaging GmbH, Göttingen, Germany) as described previously [6,40].

#### 4.4.2. Assessment of Fibrosis by Hydroxyproline (HOP) Assay

To assess the extent of fibrosis and collagen content in the hearts of mice, ~20 mg of flash-frozen heart tissue from ~9 mo-old female and male Tg-D94A and NTg mice was removed from left ventricles and boiled in 200 μL of 6 M HCl (Macron fine chemicals, Center Valley, PA, USA) at 110 °C overnight. 5 μL aliquots of hydrolyzed tissue were added to 80 μL of 100% isopropanol (Sigma-Aldrich, St. Louis, MO, USA) and allowed to react with 40 μL of 7% chloramine-T solution (Sigma-Aldrich, St. Louis, MO, USA) mixed at 1:4 ratio with acetate citrate buffer containing 0.695 M sodium acetate, 0.174 M citric acid, 0.435 M NaOH (EMD Chemicals Inc., Gibbstown, NJ, USA) and 38.5% [*v*/*v*] isopropanol) for 5 min at room temperature. Then, 0.5 mL of Ehrlich reagent (Ehrlich stock solution: 6 g of p-dimethylaminobenzaldehyde [Sigma-Aldrich, St. Louis, MO, USA] mixed with 20 mL of ethanol and 1.35 mL of sulfuric acid [Mallinckrodt Baker, Inc., Phillipsburg, NJ, USA]) was added to the mixture at a 3:13 ratio with isopropanol) and incubated at 55 °C for 30 min. Then, the mixture was placed on ice for 5 min and centrifuged at 5000× *g* for 1 min at 4 °C. 200 μL aliquots were placed in a 96-well plate and the absorbance was measured at 558 nm. The standard curve of trans-4-hydroxy-L-proline (Sigma-Aldrich, St. Louis, MO, USA) (0–1000 μM) was used to determine the total amount of hydroxyproline [HOP] in LV tissue (mg) from all tested mice [16].

#### 4.4.3. Transmission Electron Microscopy (TEM)

TEM imaging was conducted in the EM Core Facility at the University of Miami Miller School of Medicine. The anterior section of each heart from ~6-mo-old female Tg-D94A and NTg mice (previously immersed in 10% buffered formalin overnight) was fixed with 2% glutaraldehyde (Electron Microscopy Sciences, Hatfield, PA, USA), and left ventricles were sectioned longitudinally for imaging, as described previously [6,41]. The grids were examined using a JEOL JEM-1400 electron microscope (JEOL, Tokyo, Japan) equipped with an AMT BioSprint 12 digital camera (AMT Imaging, Woburn, MA, USA) at 1000×, 3000×, and 5000× magnification. The average mitochondrial area was assessed using 3000× magnification by Image J. Five to seven images for each group were used to determine the intermyofibrillar mitochondria (IFM) content.

### 4.5. Assessment of Gene Expression Profiles in the Hearts of HCQ- vs. Placebo-Treated Mice

After 30 days of treatment, total RNA (3 μg) was isolated from ventricles of male and female HCQ- and placebo-treated Tg-D94A vs. NTg and converted into double-stranded cDNA using Random Primers and a High-Capacity cDNA Reverse Transcription Kit (Applied Biosystems, Waltham, MA, USA) [42]. Quantitative qPCR was conducted using SYBR Green I chemistry with gene-specific QuantiTect Primer Sets (Qiagen, Hilden, Germany) for murine Atrial Natriuretic Factor (ANF, NM_008725), B-type natriuretic peptide (BNP, NM_008726); collagen type VIII, alpha I (Col8a1, NM_007739); Angiotensin-converting enzyme 2 (ACE2, NM_027286, and NM_001130513). Glyceraldehyde-3-phosphate dehydrogenase (GAPDH, NM_008084) was used as a housekeeping gene. Primer sets and Power SYBR Green PCR Master Mix (Applied Biosystems) were used according to the manufacturer’s protocol. All reactions were performed in duplicate and were run in Bio-Rad CFX connect Real-time system (Bio-Rad Laboratories, Hercules, CA, USA).

### 4.6. Mechanical Measurements on Skinned Papillary Muscle Fibers

#### 4.6.1. Preparation of Fibers

LVPM fibers were isolated from treated and untreated Tg-D94A and NTg hearts and dissected into small muscle bundles (2–3 mm in length and 0.5–1 mm in diameter) in ice-cold pCa 8 solution containing 10^−8^ M [Ca^2+^], 1 mM free [Mg^2+^] [total MgPr (propionate, BOC Sciences, Shirley, NY, USA) = 3.88 mM], 7 mM EGTA, 2.5 mM [Mg-ATP^2-^], 20 mM MOPS pH 7.0, 15 mM creatine phosphate, and 15 U/mL of phosphocreatine kinase (Sigma-Aldrich, St. Louis, MO, USA), ionic strength = 150 mM adjusted with KPr, substituted by 30 mM BDM and 15% glycerol. After dissection, the bundles were transferred to pCa 8 solution, mixed with 50% glycerol (storage solution), and incubated for 1 h on ice. Then, they were chemically skinned with 1% Triton X-100 (Sigma-Aldrich, St. Louis, MO, USA) and transferred to 50/50 (%) pCa 8 and glycerol overnight at 4 °C. Lastly, the bundles were transferred to a new storage solution and kept at −20 °C for 5–10 days [43].

#### 4.6.2. Assessment of Maximal Contractile Force and Force-pCa in Skinned LVPM Strips

Muscle fiber mechanics experiments were carried out on skinned LVPM fibers from mice at room temperature as described previously [6]. Briefly, small muscle strips (~1.5 mm in length and ~100 μm in diameter) were isolated from glycerinated skinned LVPM bundles, rinsed in pCa 8 solution, and attached to a Guth force transducer (Guth Muscle Research System, Heidelberg, Germany). They were then soaked in pCa 8.0 solution containing 15 units/mL of creatine phosphokinase for a few min followed by further skinning for 30 min in pCa 8 solution, which contained 1% Triton X-100. After rinsing in pCa 8 solution, their length was adjusted to remove the slack. This procedure resulted in a sarcomere length (SL) ~2.1 µm as judged by the first-order optical diffraction using HeNe laser (wavelength: 0.6328 µm). The strips were then relaxed in the pCa 8 solution, and the maximal force was determined in the pCa 4 solution (same composition as pCa 8 except [Ca^2+^] = 10^−4^ M). After pCa 4 force determination, the strips were exposed to solutions of increasing concentration of Ca^2+^ (pCa 8 to 4), and the force-pCa relationship was established. The data were plotted using the Hill equation that yielded the pCa_50_ value and the Hill coefficient (n_H_) [6].

### 4.7. Single ATP Turnover Rate Measurements in Skinned LVPM Fibers

LVPM fibers were isolated from HCQ- or placebo-treated Tg-D94A and NTg mice, chemically skinned, dissected to about 100 µm in diameter and subjected to N-methylanthraniloyl (mant)-ATP chase assays using the IonOptix instrumentation (IonOptix, LLC, Westwood, MA, USA), as described previously [30,31]. Briefly, fibers were incubated in a rigor buffer (120 mM K^+^-propionate (KPr), 5 mM MgPr, 2.5 mM K_2_HPO_4_, 2.5 mM KH_2_PO_4_, and 50 mM MOPS (Sigma-Aldrich, St. Louis, MO, USA), pH 6.8, with freshly added 2 mM DTT (Bioworld molecular life sciences, Dublin, OH, USA) containing 250 μM mant-ATP (Thermo Fisher Scientific, Waltham, MA, USA) for several minutes until maximum fluorescence was reached. Fluorescence decay curves vs. time were collected when mant-ATP was rapidly exchanged with 4 mM non-labeled ATP applied in various pCa solutions containing 10^−8^ to 10^−4^ M [Ca^2+^], 5 mM free [Mg^2+^], total MgPr = 5.69 mM, 7 mM EGTA, 50 mM MOPS, pH 6.8, and ionic strength = 170 mM (adjusted with KPr). Fluorescence decay isotherms vs. time (in seconds) were fitted to a two-state exponential equation using GraphPad 7 (GraphPad Software, San Diego, CA, USA) to derive the amplitudes of the fast (P1) and slow (P2) phases of fluorescence decay and their respective T1 and T2 lifetimes (in seconds) [30,31]. P1 and T1 represent the initial fast decay in fluorescence intensity, which comprises myosin in the DRX state and the release of nonspecifically bound mant-ATP, which is presumably also fast, while P2 and T2 are representative of the slow decrease in fluorescence intensity due to myosin in the SRX state [28]. To estimate the number of myosin heads that directly occupy the DRX vs. SRX states, the rapid phase of the fluorescence decay (P1) was corrected for the fast release of nonspecifically bound mant-ATP in the sample (0.44), and the number of heads in the SRX state calculated as P2/(1 − 0.44) as described in Hooijman et al. [28].

### 4.8. Statistical Analysis

All values are shown as means ± SD (standard deviation) or SEM (standard error of the mean). Statistically significant differences were determined using a t-test for comparisons between the two groups. If more than two groups were compared, one-way or two-way ANOVA with Sidak’s multiple comparison test was used with statistical significance depicted as * *p* < 0.05 (GraphPad Prism software version 7.0 for Windows, San Diego, CA, USA).

## Figures and Tables

**Figure 1 ijms-23-15589-f001:**
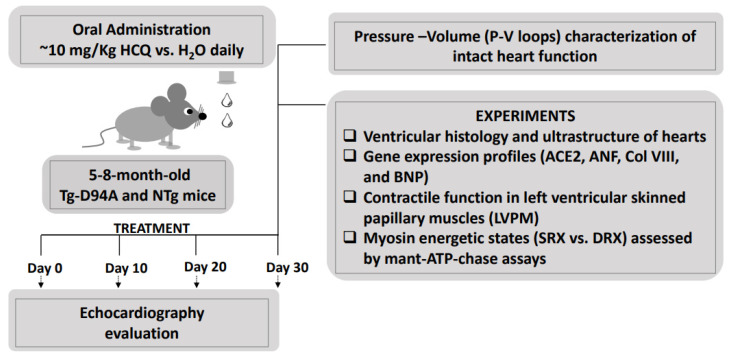
Schematic representation of the HCQ- vs. placebo (H_2_O) experimental design and the assessment of heart function in Tg-D94A vs. NTg mice. 5–8-mo-old Tg-D94A and NTg mice of both sexes were designated for HCQ (N^o^ = 16) vs. placebo (N^o^ = 16) treatment. Female and male mice were housed separately from each other, with 2–4 females and 1–5 males placed in one cage. After 30 days of treatment, the animals were evaluated for changes in the in vivo heart function, heart morphology, ultrastructure, and sarcomeric contractile and energetic properties.

**Figure 2 ijms-23-15589-f002:**
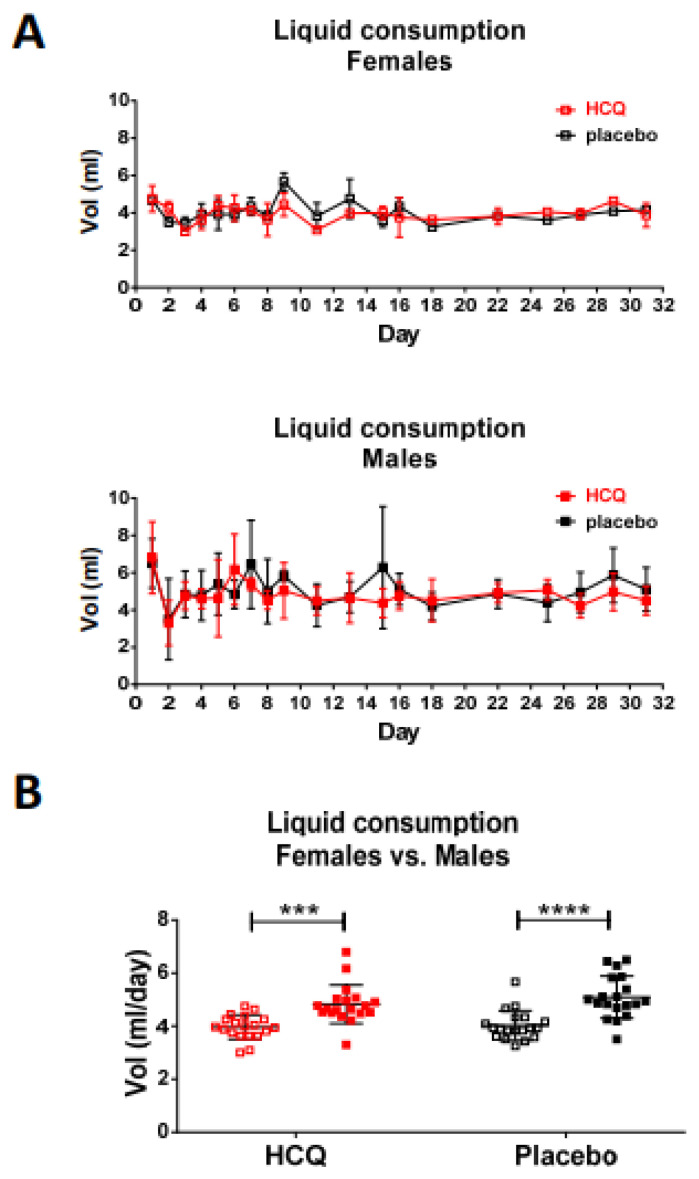
Daily fluid intake during HCQ treatment of Tg-D94A and NTg mice. (**A**) Daily liquid consumption of HCQ dissolved in drinking water (red symbols) vs. placebo/H_2_O (black symbols) by females (open symbols) and males (closed symbols). The placebo group comprised N° = 6 females and N^o^ = 6 males (3F and 3M mice per genotype), while the HCQ-treated group N^o^ = 8 females and N^o^ = 8 males (4F and 4M mice per genotype), with each gender housed in a separate cage. The volume of liquid consumption was recorded for each cage every day for the first 9 days of the experiment and then every 2–4 days, divided by the number of mice per cage and the interval between measurements, and further averaged per gender. The graphs in (**A**) show the average daily intake per animal. (**B**) Averaged liquid consumption per mouse per day. Open symbols represent female mice and closed-male mice. Data are presented as the mean ± SD. Statistical significance was assessed by two-way ANOVA followed by Sidak’s multiple comparison test and depicted as *** *p* < 0.001 and **** *p* < 0.0001 for female vs. male mice.

**Figure 3 ijms-23-15589-f003:**
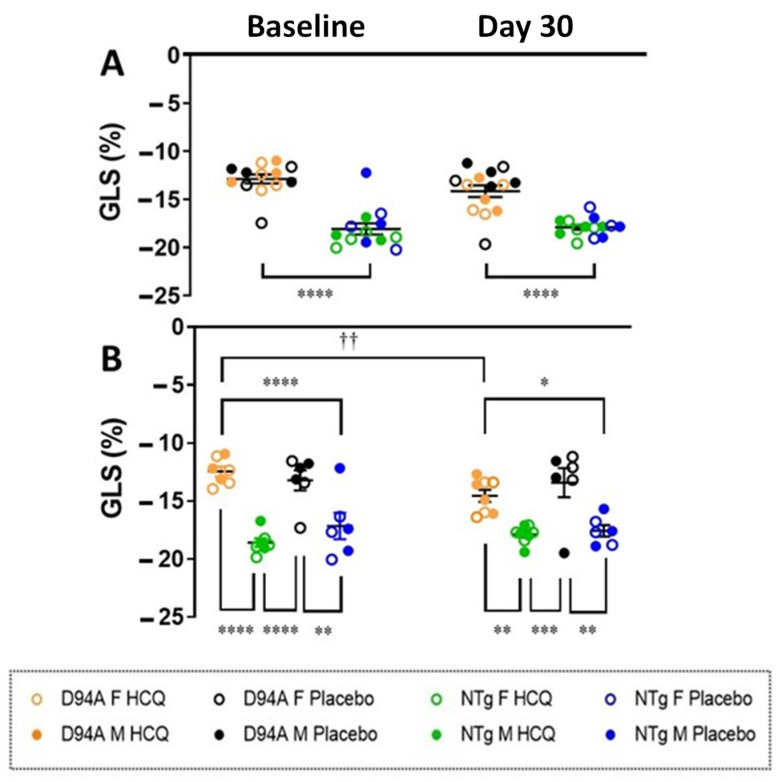
Cardiac performance of Tg-D94A vs. NTg mice assessed by speckle-tracking echocardiography. **Top** panel (**A**) shows pooled values of global longitudinal strain (GLS) from Tg-D94A vs. NTg groups at baseline and 30 days after treatment with HCQ or placebo (H_2_O), indicating an impaired cardiac function in the Tg-D94A group. The **bottom** panel (**B**) depicts GLS for each group and the effect of HCQ treatment over time. Data are presented as mean ± SEM of n = N^o^ animals (shown in Table 1), with significance calculated by unpaired *t*-test (**A**) and mixed-effects analysis followed by Sidak’s multiple comparisons test (**B**). Significance is denoted with * *p* < 0.05, ** *p* and †† *p* < 0.01, *** *p* < 0.001, and **** *p* < 0.0001. Asterisks (*) depict significant differences between Tg-D94A and NTg mice (±HCQ) and crosses (†) the differences between the values for Tg-D94A at baseline and after 30 days of HCQ treatment.

**Figure 4 ijms-23-15589-f004:**
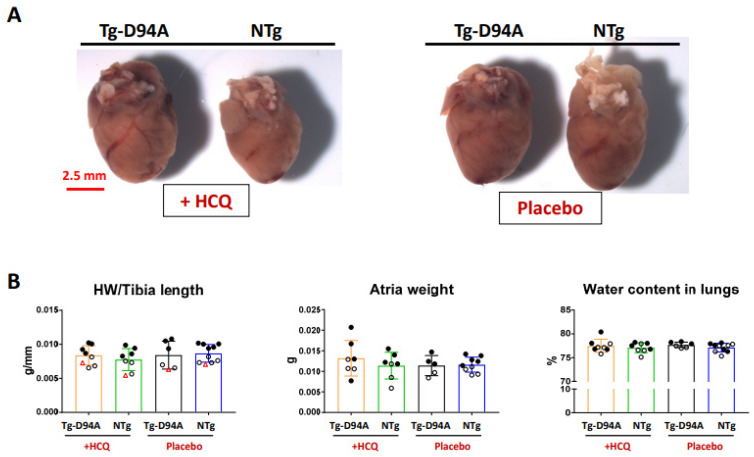
The effect of HCQ vs. placebo (H_2_O) treatment on gross morphology of Tg-D94A vs. NTg animals. (**A**) The hearts of 6-mo-old female Tg-D94A vs. NTg mice treated with HCQ vs. placebo (drinking water). (**B**) The heart weight/tibia length, atria weight, and the water content in the lungs of HCQ vs. placebo-treated Tg-D94A and NTg littermates. Hearts depicted with red triangles in the HW/tibia length graph represent the hearts pictured in Figure 4A. Open symbols show female and closed-male mice. Data are presented as the mean ± SD. No statistical significance was found by two-way ANOVA followed by Sidak’s multiple comparison test.

**Figure 5 ijms-23-15589-f005:**
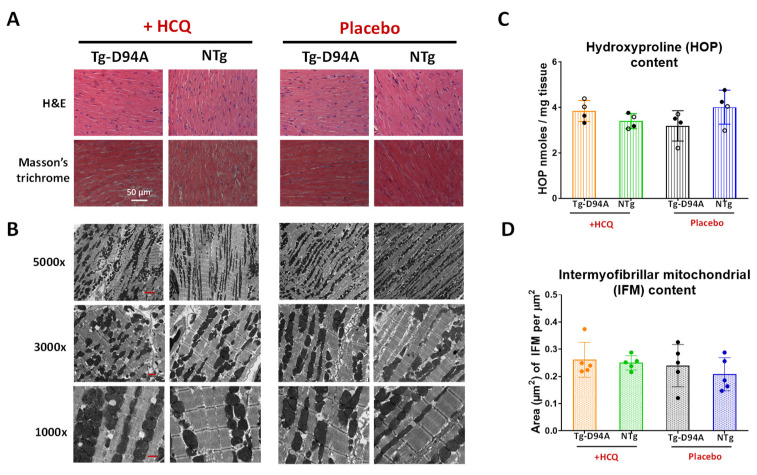
Histopathology and myocardial ultrastructure of the hearts of 6-mo-old Tg-D94A vs. NTg female mice treated with HCQ- vs. placebo. (**A**) Left ventricular (LV) samples stained with hematoxylin and eosin (H&E, **upper** panel) and Masson’s trichrome (**bottom** panel). (**B**) Transmission electron microscopy (TEM) images of sarcomeric ultrastructure of LV tissue of Tg-D94A and NTg mice. Images were taken at 1000×, 3000×, and 5000× magnification with scale bars of 4 µm, 1 µm, and 800 nm, respectively. Note some disrupted sarcomere structures and excessive vacuolar formations in the myocardium of Tg-D94A vs. NTg mice. (**C**) Quantification of fibrosis by hydroxyproline (HOP) assay. Data are presented as the mean ± SD (N^o^ = 4 animals/group). (**D**) Assessment of intermyofibrillar mitochondrial (IFM) content in the hearts of Tg-D94A vs. NTg female mice treated with HCQ- vs. placebo at 3000× magnification (scale bar 1 µm). IFM data (in µm^2^) are presented as the mean ± SD (n = 5 images). No statistical significance was found in fibrotic content or IFM between HCQ- or placebo-treated groups by two-way ANOVA followed by Sidak’s multiple comparison test.

**Figure 6 ijms-23-15589-f006:**
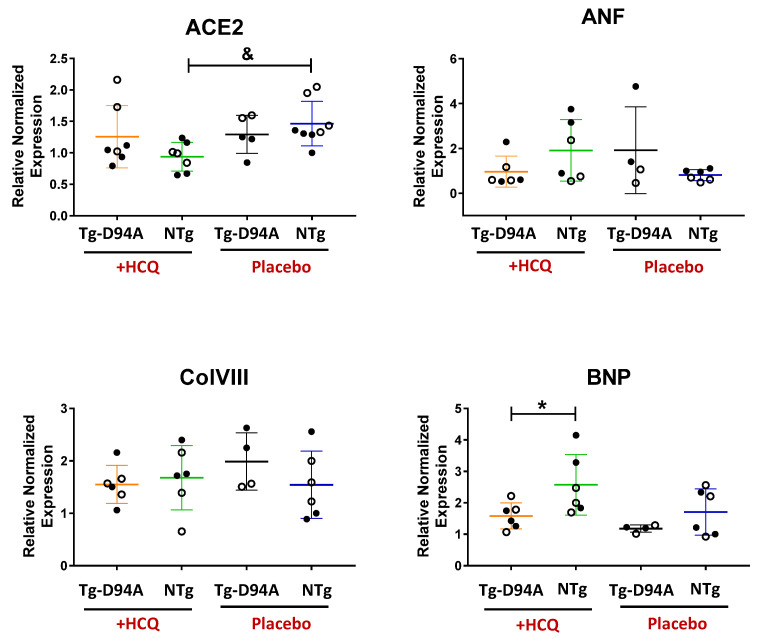
Gene expression profiles in HCQ- vs. placebo-treated Tg-D94A and NTg hearts. Expression of ACE2, ANF, BNP, and ColVIII was determined in the hearts of placebo-treated Tg-D94A vs. NTg mice and presented as relative expression normalized to the level of placebo-treated male NTg mouse (expression = 1). Data are the mean ± SD of n = N^o^ animals. N^o^ = 6–7 for HCQ-treated Tg-D94A and N^o^ = 4–5 for placebo-treated Tg-D94A. N^o^ = 6–7 for HCQ-treated NTg mice and N^o^ = 6–8 for placebo-treated NTg mice. Females are depicted by open symbols and males by closed symbols. Statistical analysis was calculated by two-way ANOVA followed by Sidak’s multiple comparisons test with significance depicted as * *p* < 0.05 for HCQ-treated Tg-D94A vs. NTg mice and ^&^
*p* < 0.05 for HCQ- vs. placebo-treated NTg mice.

**Figure 7 ijms-23-15589-f007:**
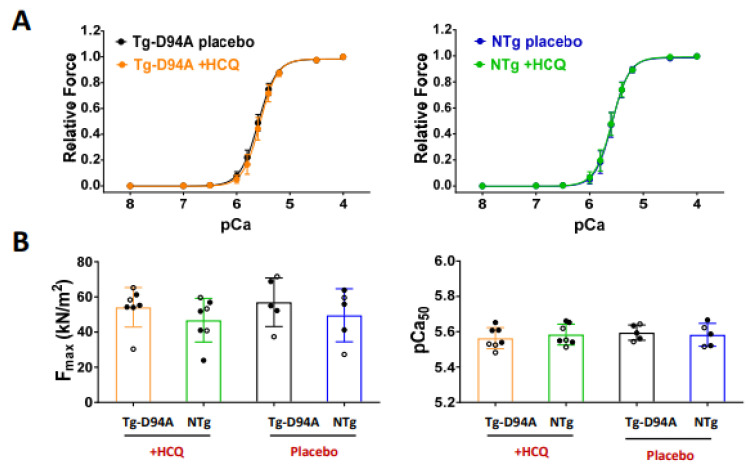
Contractile function in HCQ- vs. placebo-treated Tg-D94A and NTg hearts. (**A**) Force-pCa relationship in skinned LVPM fibers of HCQ and placebo-treated Tg-D94A mice (**left**) vs. NTg littermates (**right**). (**B**) **Left**: Maximal contractile force (in kN/m^2^) developed by LVPM fibers at saturating calcium concentrations (pCa 4) and calculated per cross-section of muscle fiber. **Right**: Myofilament calcium sensitivity of force (pCa_50_). Females are depicted with open symbols, and males are depicted with closed symbols. Each point represents an average of 2 to 4 fibers per animal. Values are the means ± SD of n = N^o^ animals. N^o^ = 7 mice for HCQ-treated Tg-D94A and N^o^ = 5 mice for placebo-treated Tg-D94A. N^o^ = 7 mice for HCQ-treated NTg and N^o^ = 5 mice for placebo-treated NTg. No statistical significance was found by two-way ANOVA followed by Sidak’s multiple comparison test.

**Figure 8 ijms-23-15589-f008:**
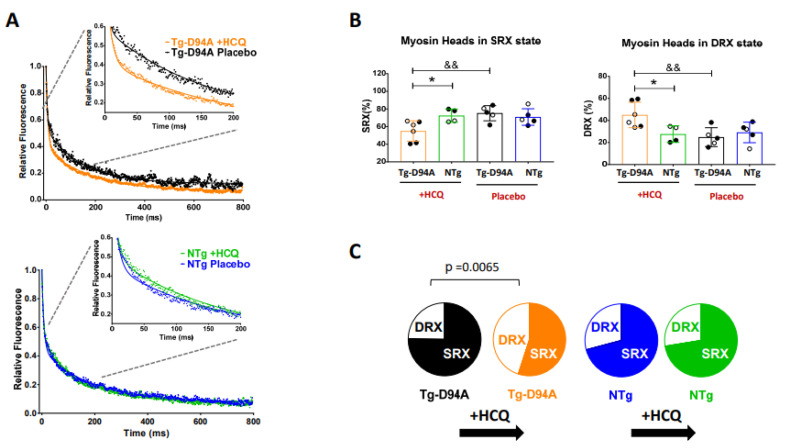
Effect of HCQ treatment on the distribution of myosin energetic states in Tg-D94A vs. NTg hearts. (**A**) Mant-ATP turnover studies and the comparison of fluorescence decay curves between HCQ- vs. placebo-treated Tg-D94A (**upper** panel) and between HCQ- vs. placebo-treated NTg mice (**bottom** panel). (**B**) Comparison of myosin heads in the SRX state (**left** panel) and DRX state (**right** panel). LVPM fibers from HCQ-treated Tg-D94A (3 males, 3 females) were compared with HCQ-treated NTg (2 males, 2 females), and placebo-treated Tg-D94A (3 males, 2 females) were compared with placebo-treated NTg (3 males, 2 females). Open symbols depict female mice, and closed symbols depict male mice. Note that % SRX is decreased in HCQ-treated Tg-D94A vs. NTg mice, and this effect is coupled with increased DRX in HCQ-treated Tg-D94A vs. NTg mice. Significant changes were noted in the SRX/RDX distribution for HCQ vs. placebo-treated Tg-D94A animals. Data points represent averaged values from 11–17 fibers/per heart and are expressed as the mean ± SD of n = N^o^ animals with significance calculated using two-way ANOVA followed by Sidak’s multiple comparison test. * *p* < 0.05 for HCQ-treated Tg-D94A vs. NTg mice and ^&&^
*p* < 0.01 for HCQ- vs. placebo-treated Tg-D94A mice. (**C**) Pie plots representing the effect of HCQ on the distribution of DRX vs. SRX states in Tg-D94A and NTg hearts. Note the significant effect of HCQ causing a decrease in the number of myosin heads occupying the SRX state while increasing the DRX heads in Tg-D94A mice (*p* = 0.0065). No effect of HCQ was observed in NTg mice.

**Table 1 ijms-23-15589-t001:** Doppler echocardiographic evaluation of Tg-D94A and NTg mice at baseline and 30 days after treatment with HCQ or placebo.

	BASELINE		DAY 30			
			HCQ		Placebo (H_2_O)	
Parameter	Tg-D94A	NTg	Tg-D94A	NTg	Tg-D94A	NTg
N^o^ of animals (M, F)	13 (6, 7)	13 (6, 7)	8 (4, 4)	8 (4, 4)	6 (3, 3)	10 (5, 5)
M-mode						
HR (bpm)	547 ± 11	538 ±13	534 ± 12.3	557 ± 13.0	543 ± 16.0	511 ± 10.3
LVID; s (mm)	2.8 ± 0.1	2.8 ± 0.1	2.9 ± 0.1	2.8 ± 0.1	2.9 ± 0.1	3.0 ± 0.1
LVID; d (mm)	4.0 ± 0.1	4.0 ± 0.1	4.0 ± 0.1	4.0 ± 0.1	4.1 ± 0.1	4.2 ± 0.1
FS (%)	29.4 ± 1.1	29.4 ± 0.9	28.3 ± 1.5	29.9 ± 1.3	29.5 ± 1.1	28.6 ± 1.1
LV Mass (mg)	107 ± 6.6	101 ± 6.4	107 ± 6.1	113 ± 10.0	116 ± 10.2	111 ± 10.7
LVAW; s (mm)	1.4 ± 0.04	1.3 ± 0.03	1.4 ± 0.07	1.4 ± 0.05	1.4 ± 0.03	1.3 ± 0.06
LVAW; d (mm)	0.9 ± 0.03	0.9 ± 0.04	0.9 ± 0.04	1.0 ± 0.04	0.9 ± 0.05	1.0 ± 0.06
LVPW; s (mm)	1.2 ± 0.03	1.1 ± 0.02	1.2 ± 0.03	1.2 ± 0.04	1.2 ± 0.05	1.1 ± 0.05
LVPW; d (mm)	0.8 ± 0.03	0.7 ± 0.02	0.8 ± 0.02	0.8 ± 0.05	0.9 ± 0.04	0.8 ± 0.03
Anatomical M-mode						
TAPSE (mm)	0.7 ± 0.03 *	0.8 ± 0.03	0.8 ± 0.03 **	0.7 ± 0.05	0.6 ± 0.03	0.7 ± 0.02
B-mode						
HR (bpm)	534 ± 8.8	532 ± 9.6	543 ± 16.3	552 ± 11.8	536 ± 12.5	514 ± 14.1
ESV (μL)	23.7 ± 1.7	26.0 ± 1.9	26.1 ± 2.0	28.5 ± 2.3	25.9 ± 2.6	29.3 ± 2.1
EDV (μL)	53.5 ± 2.4	57.4 ± 3.3	56.4 ± 3.8	56.7 ± 4.5	56.6 ± 4.6	61.4 ± 3.1
SV (μL)	30 ± 1.2	31.5 ± 1.6	30.3 ± 1.9	30.4 ± 2.0	30.7 ± 2.2	32.1 ± 1.4
EF (%)	56 ± 1.7	55 ± 0.9	53.9 ± 0.8	54.2 ± 1.1	54.6 ± 1.4	52.6 ± 1.5
CO (mL/min)	15.9 ± 0.6	16.7 ± 0.9	16.4 ± 0.9	16.7 ± 0.9	16.5 ± 1.4	16.5 ± 0.8
GLS (%)	−12.8 ± 0.5 ****††	−17.9 ± 0.3	−14.6 ± 0.5 **	−17.9 ± 0.3	−13.4 ± 1.3 **	−17.6 ± 0.5

Data are presented as mean ± SEM of n = N^o^ of animals with significance calculated by mixed-effects analysis followed by Sidak’s multiple comparisons test (* *p* < 0.05, ** *p* < 0.01, and **** *p* < 0.0001 for Tg-D94A vs. NTg (± HCQ) and †† *p* < 0.01 for Tg-D94A at baseline vs. Tg-D94A on day 30 of HCQ treatment. Asterisks (*) depict significant differences between Tg-D94A and NTg mice (±HCQ), and crosses (†) the differences between the values for Tg-D94A at baseline and after 30 days of HCQ treatment. Abbreviations: HR, heart rate; LVID, left ventricular (LV) inner diameter in systole (s) and diastole (d); FS, fractional shortening; LVAW, LV anterior wall thickness; LVPW, LV posterior wall thickness; TAPSE, tricuspid annular plane systolic excursion; ESV, LV end-systolic volume; EDV, LV end-diastolic volume; SV, stroke volume; CO, cardiac output; EF, ejection fraction.

**Table 2 ijms-23-15589-t002:** Contractile function in skinned LVPM fibers from HCQ- and placebo-treated Tg-D94A and NTg mice.

	HCQ		Placebo	
Parameter	Tg-D94A	NTg	Tg-D94A	NTg
N^°^ mice (M, F) N^°^ fibers	7 (4, 3) 16	7 (4, 3) 14	5 (3, 2) 10	5 (3, 2) 10
Fmax (kN/m^2^)	54.12 ± 4.25	46.78 ± 4.70	57.02 ± 6.19	49.57 ± 6.74
pCa_50_	5.56 ± 0.02	5.58 ± 0.02	5.60 ± 0.02	5.58 ± 0.03
n_H_	2.80 ± 0.06	2.73 ± 0.08	2.60 ± 0.12	2.89 ± 0.07

Data are the mean ± SEM of N^o^ = 5–7 mice (10–16 fibers) per group. No statistical significance was found by two-way ANOVA followed by Sidak’s multiple comparison test. Abbreviations: Male (M), female (F) mice, Fmax, maximal pCa4 force, pCa_50_, calcium sensitivity, n_H_, Hill coefficient.

**Table 3 ijms-23-15589-t003:** Effect of HCQ treatment on myosin energetic states in Tg-D94A vs. NTg hearts.

	HCQ		Placebo	
Parameter	Tg-D94A	NTg	Tg-D94A	NTg
N^o^ mice (M, F) N^o^ fibers	6 (3, 3) 17	4 (2, 2) 11	5 (3, 2) 12	5 (3, 2) 16
DRX (%)	45 ± 5 ^&&,^ *	28 ± 4	25 ± 4	29 ± 4
SRX (%)	55 ± 5 ^&&,^ *	72 ± 4	75 ± 4	71 ± 4
T1 (s)	10 ± 3	8 ± 1	6 ± 2	5 ± 1
T2 (s)	240 ± 3	207 ± 4	231 ± 8	198 ± 2

The super-relaxed state of myosin was measured by mant-ATP/ATP chase assays in LVPM fibers (11–17 per group) from the hearts of 6–9-mo-old HCQ- and placebo-treated male (M) and female (F) Tg-D94A and NTg mice. DRX and SRX states depict the % of myosin cross-bridges occupying the fast and slow ATP turnover states, respectively, with T1 and T2 representing their lifetimes (in seconds). Data are the mean ± SEM for n = N^o^ animals with significance calculated by two-way ANOVA followed by Sidak’s multiple comparison test. ^&&^
*p* < 0.01 for Tg-D94A mice treated with HCQ vs. placebo and * *p* < 0.05 for Tg-D94A vs. NTg mice treated with HCQ.

## Data Availability

Not applicable.

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
