# Peer review of "Hydroxychloroquine Mitigates Dilated Cardiomyopathy Phenotype in Transgenic D94A Mice"

_ijms, 2022, doi:10.3390/ijms232415589_

Round 1
Reviewer 1 Report
The author aimed to evaluate the effect of short-term and low-dose HCQ treatment in MYL2-associated dilated cardiomyopathy (DCM). They found that HCQ exerts no harmful effects on the heart function of NTg or DCM mice, while 30-day treatment with low-dose HCQ resulted in significant improvement in global longitudinal strain (GLS) in DCM-D94A hearts.
1. However, except the GLS analysis, most of the echocardiography and histology results did not reveal DCM-like cardiac dysfunction in Tg-D94A mice as noted in reference19 verse to NTg mice. Given the importance of the observations related to DCM, it would be important to show the expression of D94A-RLC mutant in Tg mice.
2. Rigor of data interpretation should be enhanced. This includes careful analysis of the significance of the small differences noted in several cases (e.g., Table. 1 – the notes of ** are confused.).
3. Quantitative analysis of Masson staining and EM data data is otherwise missing (e.g., quantitative analysis of, Fig 5., is critical, as "representative images" can be misleading, especially as ultrastructure of mitochondria shape and other varies considerably by one heart region to another.)”
4. The difference in the regulation of gene expression profiles in HCQ-treated Tg-D94A and NTg hearts should be discussed based on the literature and the current study. Are the protein levels and enzyme activations of ACE2 elevated in this study?
Author Response
The response to Reviewer 1 is attached.

Reviewer 2 Report
Kanashiro-Takeuchi and coworkers performed an interesting investigation aiming at demonstrating the effect of HCQ on a experimental model of DCM. Using a wide range of techniques, authors concluded that HCQ exerts no harmful effects on the heart function of NTg or DCM mice. On the contrary, 30-day treatment with low-dose HCQ resulted in significant improvement DCM hearts. However, we have further concerns about this study:
- Title: After reviewing the article we are not sure what it is exactly the aim of the manuscript. First option: HCQ is cardiotoxic in DCM hearts? Second option: HCQ could be use as a treatment for DMC? Please explain in the title the exact objective of this manuscript..
- Introduction:
· the first two paragraphs are focused on the relationship of COVID-19 and HCQ. In my opinion, it is not necessary to report the implication of this drug on COVID-19 scenario since we are in a complete different pathology (HCQ). In case the object is to test the cardiotoxicity of HCQ for COVID-19 patients, then it is necessary to explain COVID-19 situation.
· Moreover, the last part of the introduction looks like it belongs to methods section because a description of the different methodological approaches are performed. Overall, I think that the introduction should be re-write with the objective of clearly explaining what is DCM as well as previous data or evidence for testing HCQ on this situation. Lastly, authors should also include a paragraph explaining the main objectives of this work.
- Results:
· Line 156-157: it seems this sentence belongs to discussion instead of results.
· Figure 3: please indicate what means asterisk and cross.
· Table 1 and Table 2 provide a huge amount of data and some of the parameters are not necessary to understand the results. I would suggest to reduce the number of variables in both tables to make it clearer and move to supplementary materials those that are not crucial/significant.
· Since males have a higher liquid consumption, it would mean that they took a higher dose of HCQ. Did you evaluate whether there are any differences in the results depending on animal sex?
· In Fig 6, I was wondering why you chose these four genes to determine their mRNA expression? It is surprising that no differences were detected between the control vs DCM groups.
· Please indicate the number of animals you use in each experimental group.
- Discussion:
· Please remove all references to COVID-19 because it does not fit with the main objective of this article (depending on the objective of your work).
· It would be great to add a paragraph about potential clinical applications to these results.
· Some of the obtained results are negative or does not provide new insight into DCM pathophysiology. For that reason, I think results need further discussion and comparison to previously published articles.
MINOR CHANGES
· I would rather suggest not to include abbreviations in the title.
· Figure legends: please try to homogenize the way you refer to panel. Sometimes are in bold with parenthesis or without them.
Author Response
The response to Reviewer 2 is attached

Round 2
Reviewer 1 Report
The manuscript aimed to investigated whether short-term and low-dose treatment HCQ
with can modulate heart function in a preclinical model of dilated cardiomyopathy. Although the study results may be of interest, there are some concerns that should be solved.
1 The revised manuscript is too messy to read.
2 It seems that the dilated function measured echocardiography was not obvious.
3 The authors should measure the protein levels by WB.
Reviewer 2 Report
The authors have made all the modifications I have previously asked, so the manuscript is suitable for publication in the current form.
Round 3
Reviewer 1 Report
No comments further.